# AdaLomo: Low-memory Optimization with Adaptive Learning Rate

## Abstract

Large language models have achieved remarkable success, but their extensive parameter size necessitates substantial memory for training, thereby setting a high threshold. While the recently proposed low-memory optimization (LOMO) reduces memory footprint, its optimization technique, akin to stochastic gradient descent, is sensitive to hyper-parameters and exhibits suboptimal convergence, failing to match the performance of the prevailing optimizer for large language models, AdamW. Through empirical analysis of the Adam optimizer, we found that, compared to momentum, the adaptive learning rate is more critical for bridging the gap. Building on this insight, we introduce the low-memory optimization with adaptive learning rate (AdaLomo), which offers an adaptive learning rate for each parameter. To maintain memory efficiency, we employ non-negative matrix factorization for the second-order moment estimation in the optimizer state. Additionally, we suggest the use of a grouped update normalization to stabilize convergence. Our experiments with instruction-tuning and further pre-training demonstrate that AdaLomo achieves results on par with AdamW, while significantly reducing memory requirements, thereby lowering the hardware barrier to training large language models.

## 1    Introduction

Large language models (Scao et al., 2022; Zhang et al., 2022; Touvron et al., 2023a;b) have garnered increasing attention due to their exceptional capabilities across a diverse range of tasks. Either supervised fine-tuning or further pre-training can lead to enhanced performance. As the number of parameters grows, the substantial GPU memory required for training sets a high hardware threshold. Recently, Lv et al. (2023) has proposed low-memory optimization (LOMO) to train large language models in a memory-saving approach by simultaneously backpropagating gradients and updating parameters during the backward pass, enabling the fine-tuning of all parameters of a 7B model on a consumer-grade RTX 3090.

While LOMO's performance on the SuperGLUE (Wang et al., 2019) benchmark is comparable to popular parameter-efficient fine-tuning methods (Ding et al., 2023; Hu et al., 2022), it falls short on a broader range of tasks against adaptive optimization methods like Adam (Kingma & Ba, 2015), exhibiting a convergence gap. We attribute this to its reliance on the naive stochastic gradient descent optimization approach. We analyze the differences in optimization methods between Adam and LOMO. Compared to LOMO, Adam incorporates both the first and second moment estimation in its optimizer state, which are the moving averages of the gradient and the squared gradient, respectively. Based on our empirical analysis, we identify that the second moment estimation is the pivotal factor influencing the convergence of training large language models between LOMO and Adam.

The second-order moment estimation in Adam serves to offer an adaptive learning rate for each parameter. Expanding on this concept, we introduce the low-memory optimization with adaptive learning rate (AdaLomo), which similarly provides an adaptive learning rate for each parameter. To retain memory efficiency, inspired by Adafactor (Shazeer & Stern, 2018), we employ non-negative matrix factorization (Yu et al., 2018) for the second-order moment estimation in the optimizer state. Furthermore, we advocate for the use of a grouped update normalization to stabilize convergence. The number of trainable parameters and the GPU memory consumption for model state under mixed-

Table 1: Trainable parameter number and memory usage under mixed-precision training. $N \ll M$ and $O(M + N) = O(M)$, where $M$ is the number of model parameters. AdaLomo's memory consumption is comparable to LoRA, and its trainable parameter number is equivalent to AdamW.

| Method | Trainable Params (Billion) | Memory (GB) | | | |
|---|---|---|---|---|---|
| | | Param | Gradient | Optimizer State | Total |
| LoRA | $N$ | $2M$ | $O(N)$ | $O(N)$ | $\sim 2M$ |
| AdamW | $M$ | $2M$ | $2M$ | $12M$ | $16M$ |
| AdaLomo | $M$ | $2M$ | $O(N)$ | $O(N)$ | $\sim 2M$ |

precision training among AdaLomo, the popular LoRA (Hu et al., 2022) method, and the AdamW optimizer (Loshchilov & Hutter, 2019) are compared in Table 1. Our contributions are as follows:

1. We examined the distinctions between the LOMO and Adam optimization techniques. Empirical analysis revealed that the primary difference in performance between LOMO and Adam, especially when training large language models, stems from Adam's incorporation of second-moment estimation to furnish an adaptive learning rate for each parameter.

2. We introduce AdaLomo, which provides an adaptive learning rate for each parameter while maintaining memory efficiency, democratizing the training of large language models. In AdaLomo, we also employ grouped update normalization to stabilize the training process.

3. We evaluate the performance of large language models post instruction-tuning with AdaLomo across five benchmarks spanning diverse tasks. The results are comparable to both AdamW and LoRA. Furthermore, when AdaLomo is used for additional pre-training on Chinese and Python code, its performance is on par with that of AdamW.

4. We profile the memory consumption and throughput of AdaLomo. Its reduced memory usage and reasonable computational overhead make it a viable option for training large language models.

## 2 PRELIMINARIES

In the subsequent sections of this paper, we use $\boldsymbol{\theta_t}$ to denote the parameters of the model at the $t^{th}$ step of the training process. $\theta_{t,i}$ represents the parameter at the $i^{th}$ gradient computation during the backpropagation process of $\boldsymbol{\theta_t}$. We use $\boldsymbol{g_t}$ to represent the gradient of $\boldsymbol{\theta_t}$, and $g_{t,i}$ to denote the gradient of $\theta_{t,i}$. The first and second moment estimation at the $t^{th}$ training step, which are the moving averages of the gradient and the square of the gradient respectively, are represented by $\boldsymbol{m_t}$ and $\boldsymbol{v_t}$. The symbol $\alpha$ represents the learning rate.

### 2.1 FUSED BACKWARD

In the training process, the memory is primarily consumed by the optimizer states, parameters, and gradients. The fused backward proposed in LOMO refers to the process that simultaneously calculates gradients and updates parameters during backpropagation. This can effectively reduce the memory consumption of gradients.

For a given parameter $\theta_{t,i}$, its gradient $g_{t,i}$ resides in the GPU memory until the gradient $g_{t,i+1}$ corresponding to the subsequent parameter $\theta_{t,i+1}$ is computed. Subsequently, LOMO utilizes a standard gradient descent approach for parameter updates, as depicted by the following equation:

$$\theta_{t,i} = \theta_{t-1,i} - \alpha \times g_{t,i}. \tag{1}$$

For transformer-based language models, $g_{t,i}$ is unnecessary in subsequent backpropagation steps and can be eliminated from memory. Consequently, at any given moment, the memory retains the gradients of only two consecutive parameters. The memory usage for gradients remains constant regardless of the language model's scale, yielding an $O(1)$ memory footprint. In the case of large language models, such as LLaMA-65B (Touvron et al., 2023a) with its 82 layers and 723 weight matrices, the memory consumption for gradients becomes negligible compared to that for parameters or optimizer states.

**Gradient Normalization**   Gradient normalization is instrumental in mitigating gradient vanishing and explosion. Nevertheless, its integration into LOMO presents challenges. Specifically, gradient normalization necessitates the computation of a scaling factor derived from the gradients of all parameters. This factor subsequently informs parameter updates. In the context of LOMO, however, the gradients for all parameters have not yet been computed. To incorporate gradient normalization within LOMO, two backward passes are essential:

1. Execute the first backward pass without updating the parameters and record the sum of the squares of all parameter gradients.
2. Based on the recorded values above, compute the overall gradient scaling factor.
3. Conduct the second fused backward pass, updating the parameters using the derived scaling factor.

## 2.2   EMPIRICAL ANALYSIS ON THE TWO MOMENTS IN ADAM

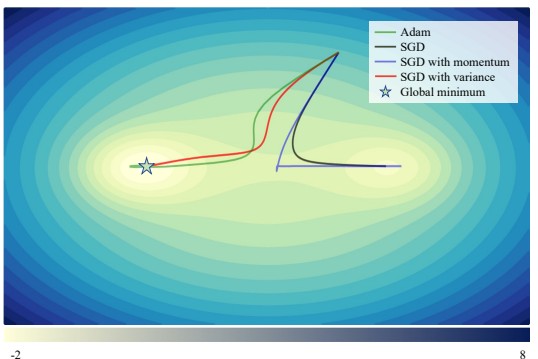

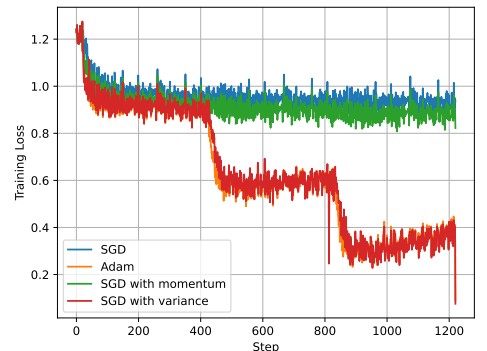

(a) Loss trajectories of different optimizers starting from the same initial point. Both Adam and SGD with variance converge to the global optimum on the left, while SGD and SGD with momentum converge to the local optimum on the right.

(b) Training loss curves of different optimizers on the same dataset. Both Adam and SGD with variance exhibit a stepwise decline in loss as the training epochs increase, ultimately achieving a lower loss than both SGD and SGD with momentum.

Figure 1: Empirical analysis on different optimizers.

LOMO exhibits efficient memory usage, essentially pushing the optimization of large language models with gradients to the extreme. However, the naive gradient descent method shown in Equation 1 faces challenges such as the propensity to get trapped in saddle points and sensitivity to the learning rate (Dauphin et al., 2014; Darken et al., 1992). Building upon SGD, a series of advanced optimization methods have been proposed that have been proven both theoretically and practically to address these challenges (Ruder, 2016). These methods typically introduce additional optimizer states, such as momentum (Qian, 1999), Nesterov accelerated gradient (Nesterov, 1983), and moving averages of squared past gradients (Duchi et al., 2011; Zeiler, 2012; Kingma & Ba, 2015), leading to extra memory consumption. Among these, the Adam series of optimizers are most widely used in training large language models, simultaneously incorporating first-moment ($m_t$) and second-moment ($v_t$) estimation for parameter updates, as demonstrated in the following equation,

$$
\begin{cases}
m_t = \beta_1 m_{t-1} + (1 - \beta_1) g_t & , \\
v_t = \beta_2 v_{t-1} + (1 - \beta_2) g_t^2 & , \\
\hat{m}_t = \dfrac{m_t}{1 - \beta_1^t} & , \\
\hat{v}_t = \dfrac{v_t}{1 - \beta_2^t} & , \\
\theta_t = \theta_{t-1} - \alpha \dfrac{\hat{m}_t}{\sqrt{\hat{v}_t} + \epsilon} & ,
\end{cases}
\tag{2}
$$

where $\epsilon$ is a small quantity introduced to prevent division by zero in calculations. The hyper-parameters $\beta_1, \beta_2 \in [0, 1)$ dictate the exponential decay rates of the respective moving averages.

We empirically investigated the differences in convergence behaviors between Adam and SGD under the function $f(x, y) = x^2 + y^2 - 2e^{-5[(x-1)^2 + y^2]} - 3e^{-5[(x+1)^2 + y^2]}$ and the fine-tuning of large language models. To ablatively analyze the roles of the first and second moments of the gradients in Adam, we conducted experiments retaining only the first-order moment estimate or the second-order moment estimation in Adam, respectively. The update rule retaining only the first-order moment estimation (or momentum) is:

$$\begin{cases} \boldsymbol{m}_t = \beta_1 \boldsymbol{m}_{t-1} + (1 - \beta_1)\boldsymbol{g}_t & , \\ \hat{\boldsymbol{m}}_t = \dfrac{\boldsymbol{m}_t}{1 - \beta_1^t} & , \\ \boldsymbol{\theta}_t = \boldsymbol{\theta}_{t-1} - \alpha \times \hat{\boldsymbol{m}}_t & . \end{cases} \quad (3)$$

Meanwhile, the update rule retaining only the second-order moment estimation (or variance) is:

$$\begin{cases} \boldsymbol{v}_t = \beta_2 \boldsymbol{v}_{t-1} + (1 - \beta_2)\boldsymbol{g}_t^2 & , \\ \hat{\boldsymbol{v}}_t = \dfrac{\boldsymbol{v}_t}{1 - \beta_2^t} & , \\ \boldsymbol{\theta}_t = \boldsymbol{\theta}_{t-1} - \alpha \dfrac{\boldsymbol{g}_t}{\sqrt{\hat{\boldsymbol{v}}_t} + \epsilon} & . \end{cases} \quad (4)$$

The results of the convergence analysis are shown in Figure 1. In Figure 1a, starting from the same initial point, Adam converges to the global optimum while SGD gets trapped at a local optimum. In the instruction-tuning scenario depicted in Figure 1b, we trained LLaMA-7B (Touvron et al., 2023a) with the Alpaca dataset (Taori et al., 2023; Wang et al., 2023) for three epochs. The loss curve of Adam during these three epochs exhibits a step-like decline, achieving a significantly smaller empirical loss compared to SGD.

Through our ablation study on Adam, we found that its second-order moment estimation has a significantly greater impact on its convergence than the first-order moment estimation. The second-order moment estimation is particularly effective for handling sparse data, allowing parameters that are infrequently updated to receive larger update steps.

Furthermore, the second-order moment in the optimizer's state has been proven to be decomposable or compressible to reduce memory usage. For example, Adafactor decomposes the second moment $v_{t,i} \in \mathbb{R}^{m \times n}$ by minimizing the I-divergence into $r_{t,i} \in \mathbb{R}^{m \times 1}$ and $c_{t,i} \in \mathbb{R}^{1 \times n}$ such that

$$v_{t,i} = r_{t,i} c_{t,i}. \quad (5)$$

The update formulas for $r_{t,i}$ and $c_{t,i}$ in Adafactor are as follows:

$$r_{t,i} = \beta_1 r_{t-1,i} + (1 - \beta_1)g_{t,i}^2 \mathbf{1}_n, \quad (6)$$

$$c_{t,i} = \beta_2 c_{t-1,i} + (1 - \beta_2)\mathbf{1}_m^T g_{t,i}^2, \quad (7)$$

where $\mathbf{1}_n$ and $\mathbf{1}_m^T$ are all-ones vectors of dimensions $n \times 1$ and $1 \times m$, respectively.

## 3 METHOD

In this section, we introduce our proposed memory-efficient optimization algorithm, Adalomo. This algorithm has demonstrated performance comparable to the current de facto optimization method for large language models, AdamW, requiring less memory consumption.

### 3.1 ADALOMO

Based on the analysis in Section 2.2, to achieve improved optimization while maintaining low memory consumption, we decided to incorporate a second-order moment estimation and discard the

---

**Algorithm 1** AdaLomo

---

**Require:** model $f(\cdot)$ with parameter $\boldsymbol{\theta}$, learning rate $\alpha$, max step $T$, training dataset $\mathcal{D}$, loss function $\mathcal{L}$, decay coefficient $\beta$, regularization constant $\epsilon$

1: **for** $t = 1$ **to** $T$ **do**
2:      sample batch $\mathcal{B} = (\boldsymbol{x}, \boldsymbol{y}) \subset \mathcal{D}$
3:      $\hat{\boldsymbol{y}} \leftarrow f(\boldsymbol{x}, \boldsymbol{\theta})$                                                  ▷ forward pass
4:      $\ell \leftarrow \mathcal{L}(\boldsymbol{y}, \hat{\boldsymbol{y}})$
5:      **for** each parameter $\theta_i$ in the order of backpropagation **do**
6:          $g_{t,i} = \nabla_{\theta_{t-1,i}} \ell$                           ▷ $g_{t,i-1}$ needed for computing $g_{t,i}$
7:          $r_{t,i} = \beta r_{t-1,i} + (1-\beta) g_{t,i}^2 \mathbf{1}_n$
8:          $c_{t,i} = \beta c_{t-1,i} + (1-\beta) \mathbf{1}_m^T g_{t,i}^2$
9:          $v_{t,i} = r_{t,i} s_{t,i}$
10:         $u_{t,i} = g_{t,i} / v_{t,i}$
11:         $\hat{u}_{t,i} = u_{t,i} / \max(1, RMS(u_{t,i})) \times \max(\epsilon, RMS(\theta_{t-1,i}))$
12:         $\theta_{t,i} = \theta_{t-1,i} - \alpha_t \hat{u}_{t,i}$
13:         $g_{t,i-1} \leftarrow$ None                                       ▷ clear $g_{t,i-1}$
14:      **end for**
15: **end for**

---

first-order moment. In our pursuit of further memory efficiency, we applied non-negative matrix factorization to the second-order moment, inspired by Adafactor. For each parameter $\theta_i$ within the model parameters $\boldsymbol{\theta}$, we introduce two optimizer states, $r_i$ and $c_i$. For parameters of size $m \times n$, we store only $r_i$ and $c_i$ instead of storing $v_i$. The size of the optimizer states is $m + n$, which is negligible compared to the size of the parameters.

In contrast to Adafactor, we update the optimizer state during the gradient backpropagation process. During parameter updates, we compute $v_i = r_i c_i$ using $r_i$ and $c_i$ to provide adaptive learning rate for the parameters. Compared to LOMO, this introduces additional computational overhead. Our experiments in Section 4.3 demonstrate that this added computational burden is acceptable. The details of the algorithm are presented in Algorithm 1.

### 3.2 GROUPED UPDATE NORMALIZATION

We utilize grouped update normalization in the AdaLomo update process, which entails adaptive modifications for the update of each parameter and helps maintain model stability especially during large-scale training. Grouped update normalization ensures that each parameter's update is meaningful and not overshadowed by large gradient values from other parameters, facilitating faster convergence and sustained stability. In contrast, global update normalization, where all parameters share a single scaling factor, might lead to some parameters updating too rapidly or too slowly, thereby affecting both convergence speed and stability. This is especially evident in large language models where different layers and parameters can exhibit considerable variations in gradient magnitudes, rendering global scaling potentially less effective.

As shown in line 11 of Algorithm 1, for the update matrix $u_i$ for parameter $\theta_i$, before applying it to the weight matrix, we divide it by the parameter-wise root-mean-square (RMS) of $u_i$ [1]. Additionally, we utilize the parameter-wise RMS of $\theta_i$ to ensure the update step size is proportional to the magnitude of the parameter.

Furthermore, it's worth noting that grouped update normalization integrates seamlessly with AdaLomo's fused backward process. While global update normalization requires two backward passes as gradient normalization mentioned in Section 2.1, grouped update normalization allows us to normalize the update matrices within a single fused backward pass.

---

[1] The root-mean-square (RMS) of $u$ is given by $RMS(u) = \sqrt{\frac{\sum_{i=1}^{i=n} u_i}{n}}$, where $n$ is the number of elements in $u$.

Table 2: Performance of the LLaMA series models on various benchmarks after instruction-tuning with different optimization techniques. Bolded numbers indicate the best results for models of the same size on a given benchmark. "N/A" denotes that no instruction-tuning is performed.

| Model | Method | MMLU | BBH | GSM8K | HumanEval | AlpacaFarm | Avg. |
|---|---|---|---|---|---|---|---|
| LLaMA-7B | N/A | 31.5 | 32.3 | 10.9 | 11.6 | 4.2 | 18.1 |
| | LoRA | 33.5 | 34.8 | 12.3 | 11.0 | 41.1 | 26.5 |
| | AdamW | 39.3 | 34.4 | 9.6 | 11.6 | 50.6 | 29.1 |
| | LOMO | 30.7 | 34.0 | 12.0 | **12.8** | 30.6 | 24.0 |
| | AdaLomo | **39.5** | **36.0** | **14.4** | 11.0 | **53.3** | **30.8** |
| LLaMA-13B | N/A | 45.2 | 38.5 | 19.5 | 14.0 | 5.3 | 24.5 |
| | LoRA | 48.3 | 40.3 | 20.2 | **19.5** | 49.1 | 35.5 |
| | AdamW | 49.4 | 40.2 | 21.8 | 18.9 | 61.0 | 38.2 |
| | LOMO | 44.2 | 38.9 | 21.3 | 16.5 | 38.4 | 31.8 |
| | AdaLomo | **50.0** | **41.5** | **25.3** | 18.9 | **62.9** | **39.7** |
| LLaMA-30B | N/A | 57.7 | 51.8 | 40.3 | 20.1 | 7.1 | 35.4 |
| | LoRA | 59.3 | **52.3** | 42.8 | **26.2** | 63.3 | 48.8 |
| | AdamW | 57.3 | 49.5 | 36.6 | 21.3 | 65.5 | 46.1 |
| | LOMO | 56.3 | 51.5 | 44.4 | 18.9 | 57.8 | 45.8 |
| | AdaLomo | **59.4** | 52.1 | **48.5** | 25.6 | **69.6** | **51.0** |
| LLaMA-65B | N/A | 62.4 | 58.7 | 53.9 | 20.7 | 4.7 | 40.1 |
| | LoRA | 62.7 | 58.7 | **60.5** | **32.9** | 69.6 | **56.9** |
| | AdamW | **63.0** | 57.9 | 55.3 | 28.1 | 73.1 | 55.5 |
| | LOMO | 62.1 | 56.9 | 57.6 | 28.1 | 65.2 | 54.0 |
| | AdaLomo | 62.7 | **59.0** | 59.7 | 29.9 | **73.4** | **56.9** |

## 4 EXPERIMENTS

In this section, we evaluate the efficacy of AdaLomo in instruction-tuning and further pre-training. Additionally, we assess memory usage and throughput. Experiments are performed using the LLaMA series of models, which have parameter sizes ranging from 7 billion to 65 billion.

### 4.1 INSTRUCTION TUNING

We utilized GPT-4-Alpaca (Peng et al., 2023) as the training data to fine-tune LLaMA, incorporating 52k instruction-following demonstrations generated by GPT-4 using the Alpaca method. Besides the unaltered vanilla model and LOMO, we compared LoRA and AdamW, two prevalent methods for instruction-tuning large language models, which act as strong baselines.

We evaluated the trained models across diverse tasks: knowledge-based tasks (MMLU (Hendrycks et al., 2021)), general reasoning tasks (BBH (Suzgun et al., 2023)), mathematical tasks (GSM8K (Cobbe et al., 2021)), coding tasks (HumanEval (Chen et al., 2021)), and instruction-following tasks (AlpacaFarm (Dubois et al., 2023)). For MMLU, BBH, and GSM8K, the answers are obtained by generating, and are assessed using accuracy. The HumanEval task is evaluated using pass@1. The AlpacaFarm task is assessed by comparing the win rate of responses against those from GPT-3.5 (Brown et al., 2020), as scored by GPT-4 (OpenAI, 2023). Training and evaluation are conducted using templates provided in the Alpaca repository. Detailed hyper-parameters and templates can be found in Appendix B.

The results are presented in Table 2. Compared to the vanilla model, models trained using these methods generally exhibit improved performance, especially in instruction-following capabilities. LOMO's performance on general reasoning (BBH), mathematics (GSM8K), and coding (HumanEval) tasks was comparable to that of LoRA and AdamW across all model sizes. However, its performance on knowledge-based tasks (MMLU) and instruction-following tasks (AlpacaFarm) is relatively inferior. The performance gap between LOMO and both LoRA and AdamW on these two tasks decreases as the model size increases. By incorporating the second-order moment estima-

tion, AdaLomo addresses LOMO's limitations, achieving comparable results with AdamW across various benchmarks for all model sizes.

## 4.2 FURTHER PRE-TRAINING

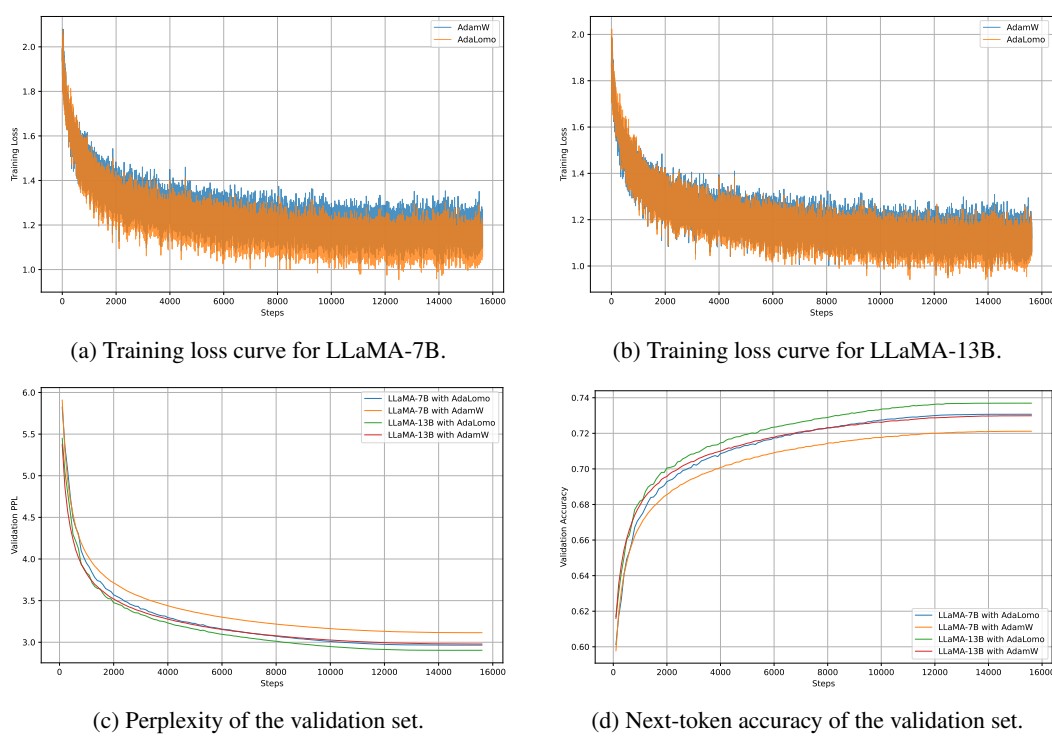

(a) Training loss curve for LLaMA-7B.

(b) Training loss curve for LLaMA-13B.

(c) Perplexity of the validation set.

(d) Next-token accuracy of the validation set.

Figure 2: Results of further pre-training in the Chinese domain.

Further pre-training refers to the additional large-scale unsupervised learning applied to a pre-trained model. We conduct further pre-training on the LLaMA model with parameter sizes of 7B and 13B in two domains: Chinese and Python code. The LLaMA model had limited exposure to data from these two domains during its initial pre-training phase. Baidu-baike is a Chinese online encyclopedia. We scraped 2 million entries from Baidu-baike for further pre-training in the Chinese domain. Additionally, we extracted 2.2 million entries from the Python subset of the StarCoder (Li et al., 2023) training dataset for further pre-training in the Python code domain. Beyond this, we set aside 2,000 entries as a validation set.

We choose AdamW as the baseline for comparison. The training hyper-parameters and data samples are detailed in Appendix C. We plot the loss curve during the model's training process and tested the perplexity and accuracy of the next-token prediction every 100 steps on the validation set.

As shown in Figure 2a and 2b, during the further pre-training in Chinese, the loss curves of AdaLomo and AdamW overlap significantly, with AdaLomo's curve slightly below that of AdamW. The fluctuation range of their losses is at a similar level. Figure 2c and 2d also indicate that AdaLomo ultimately achieved a slightly lower perplexity and accuracy on the validation set than AdamW. Both methods effectively reduced LLaMA's perplexity in Chinese, enhancing the model's performance in the Chinese domain.

Figure 3 presents the results of further pre-training in the Python code domain. The overall findings are similar to those in the Chinese domain, with some differences. Relative to Chinese, the enhancement of LLaMA's capabilities in the Python code domain through further pre-training is less pronounced. This is because, in terms of perplexity, the original LLaMA performs better on Python code than on Chinese. Although AdaLomo exhibited some fluctuations during the initial warmup phase (with a perplexity difference of less than 0.02), it converged to a more optimal point at a faster rate thereafter. The LLaMA-13B model exhibited less fluctuation than the LLaMA-7B model.

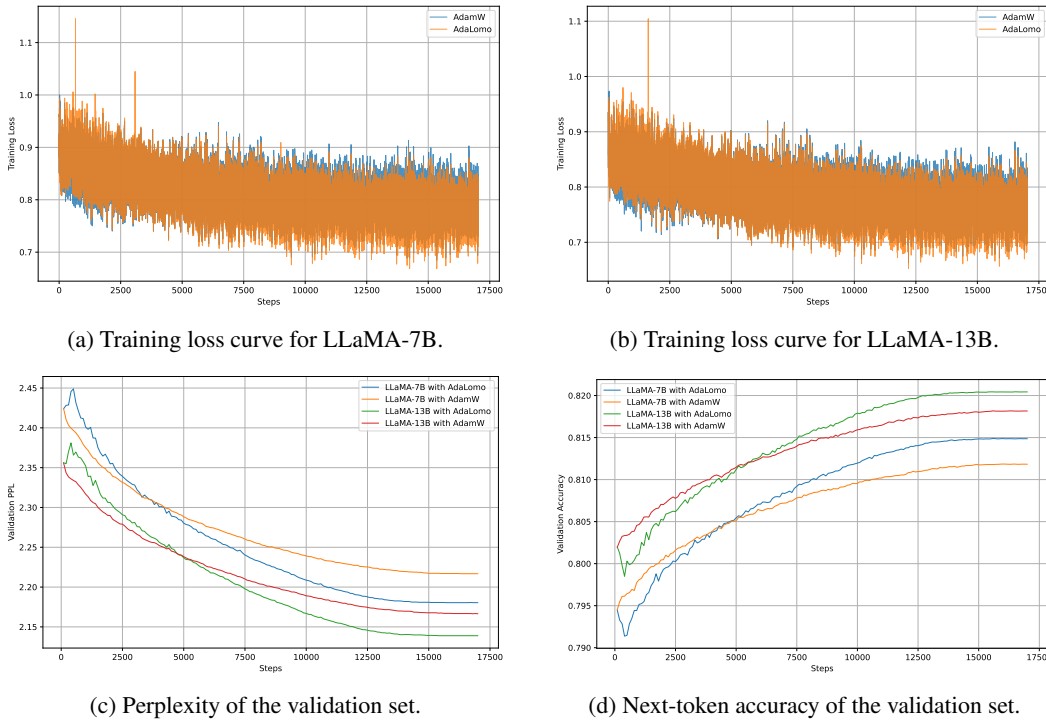

(a) Training loss curve for LLaMA-7B.

(b) Training loss curve for LLaMA-13B.

(c) Perplexity of the validation set.

(d) Next-token accuracy of the validation set.

Figure 3: Results of further pre-training in the Python code domain.

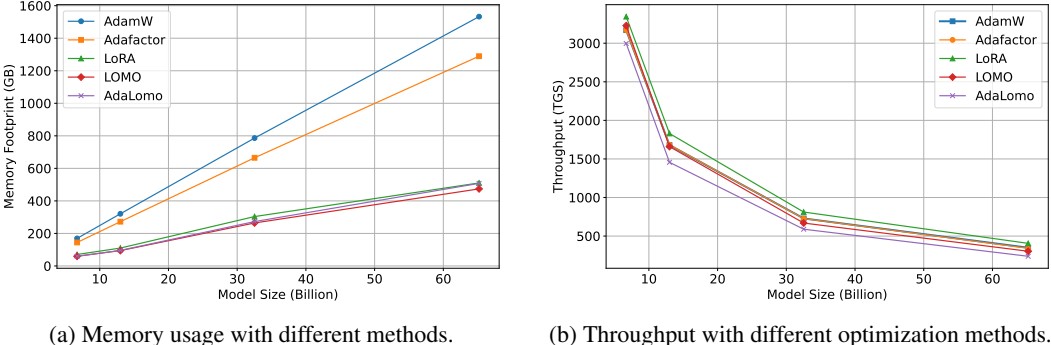

(a) Memory usage with different methods.

(b) Throughput with different optimization methods.

Figure 4: Memory footprint and throughput using different optimization methods.

We attribute these fluctuations to AdaLomo's reliance on $g_t^2$ over $v_{t-1}$ during the early stages of training, and the fact that AdaLomo does not utilize momentum.

Grouped update normalization effectively substitutes the role of gradient normalization to a certain extent. It enables stable training even without the use of gradient normalization, which is essential to prevent gradient explosion but with a decrease in throughput for LOMO. A detailed comparison regarding gradient normalization are shown in Appendix A.

### 4.3 MEMORY AND THROUGHPUT PROFILE

We evaluate the max allocated memory and throughput of AdamW, Adafactor, LoRA, LOMO, and AdaLomo, with the results in Figure 4. We employ ZeRO-3 (Rajbhandari et al., 2020) for distributed training. Throughput is measured in terms of tokens processed per GPU per second (TGS). Detailed numerical results and more specific experimental settings can be found in Appendix D.

Among the evaluated methods, AdamW exhibits the highest memory consumption. Adafactor reduces memory usage compared to AdamW by decomposing the second-order moment, resulting in memory savings proportional to the model's parameter size. AdaLomo, in comparison to LOMO, introduce an adaptive learning rate for each parameter. Nevertheless, its memory consumption remains close to that of LOMO and is comparable to LoRA, which trains with very few parameters. Due to fewer trainable parameters requiring communication during training, LoRA achieves the highest throughput. AdaLomo, which necessitates additional computations during parameter updates, shows slightly lower throughput than LOMO. All methods are tested with a consistent batch size, yet AdaLomo retains residual memory capacity, suggesting the potential for an increased batch size and greater throughput. Overall, the throughput of these methods is at the same level.

## 5 RELATED WORK

Previous research has extensively explored memory-efficient optimizers. Adafactor (Shazeer & Stern, 2018) employs non-negative matrix factorization and approximates the second-order moment estimate $v \in \mathbb{R}^{m \times n}$ using the outer product of $r \in \mathbb{R}^{m \times 1}$ and $c \in \mathbb{R}^{1 \times n}$, achieving sublinear memory consumption. The SM3 algorithm (Anil et al., 2019) introduces the cover of the parameters or, more specifically, a set of $k$ non-empty parameter groups. Each parameter is assigned an adaptive learning rate based on this cover. For a parameter matrix of size $m \times n$, the sets can be divided by rows and columns, resulting in $m + n$ sets. This reduces the memory requirement from $O(m \times n)$ to $O(m + n)$, analogous to Adafactor's memory consumption. Another line to reduce memory usage is by utilizing low-precision storage for the optimizer state. Research in DALL-E (Ramesh et al., 2021) and Gopher (Rae et al., 2021) explored the stability of 16-bit optimizers. The 8-bit Optimizer (Dettmers et al., 2022), using block-wise and dynamic exponent quantization, quantizes the optimizer states of SGDM and Adam to 8 bits. The 4-bit optimizer (Sun et al., 2020), employing the newly proposed FP4 format and the adaptive gradient scaling technique, is the first to reduce training precision to 4 bits. This results in memory consumption being reduced to one-eighth compared to 32-bit full-precision training. To decrease the memory used by gradients, LOMO updates parameters simultaneously during the gradient computation in the backward pass, a process termed as fused backward.

Additionally, there exists a series of memory-efficient optimization methods designed exclusively for fine-tuning. BBT (Sun et al., 2022b) and BBTv2 (Sun et al., 2022a) utilize evolutionary gradient-free algorithms to optimize continuous prompts without model updates. MeZO (Malladi et al., 2023) employs zeroth-order optimization methods, estimating gradients using two forward passes and optimizing the model in-place, thus equating memory consumption with inference. Parameter-efficient fine-tuning (PEFT) (Ding et al., 2023) methods selectively add or pick a subset of parameters for optimization, freezing the majority of the model parameters. For example, LoRA (Hu et al., 2022) integrates a trainable low-rank decomposition matrix into the self-attention module, thereby reducing the number of parameters to train. In comparison, AdaLomo updates all parameters using a gradient-based method, suitable for both pre-training and fine-tuning, with memory consumption comparable to PEFT methods.

## 6 CONCLUSION

In this paper, we introduce AdaLomo, designed to reduce the training barriers for large language models. By incorporating an adaptive learning rate and utilizing grouped update normalization, AdaLomo achieves results comparable to AdamW in both instruction-tuning and further pre-training. Concurrently, the memory footprint of AdaLomo is on par with the PEFT methods.

While AdaLomo is memory-efficient when training large language models, it primarily reduces the memory occupied by gradients and the optimizer states. Therefore, for models with a significant amount of activation values occupying memory, the reduction in memory usage by employing AdaLomo is limited. Thus, AdaLomo is best suited for training models with a large number of parameters. Additionally, while our experiments show that the throughput decrease is minimal, AdaLomo introduce some extra computational overhead, suggesting a direction for further improvement. This framework can be extended to optimizers using other update methods, such as SM3, and can also be adapted to methods related to optimizer states compression.

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

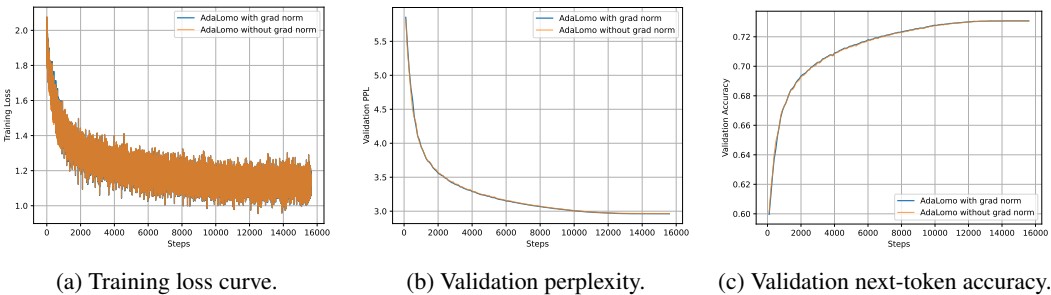

(a) Training loss curve.     (b) Validation perplexity.     (c) Validation next-token accuracy.

Figure 5: Results of further pre-training of LLaMA-7B with AdaLomo in the Chinese domain with and without gradient normalization.

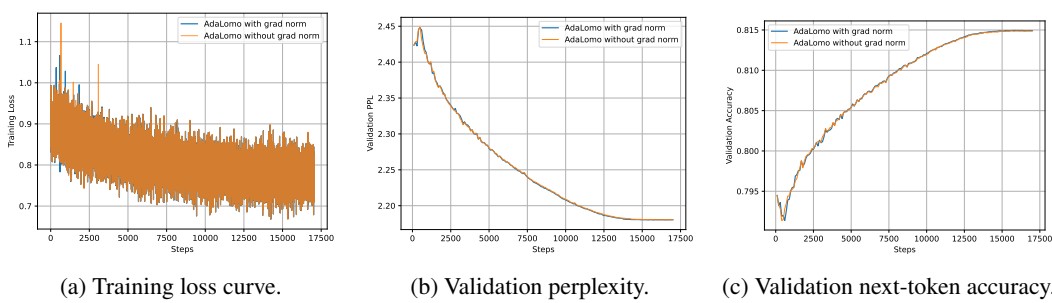

(a) Training loss curve.     (b) Validation perplexity.     (c) Validation next-token accuracy.

Figure 6: Results of further pre-training of LLaMA-7B with AdaLomo in the Python code domain with and without gradient normalization.

## A  GRADIENT NORMALIZATION FOR ADALOMO

We conduct experiments on the LLaMA-7B to assess the effects of using gradient normalization during the further pre-training of AdaLomo. Comparative experiments in the Chinese domain are illustrated in Figure 5, while those in the Python code domain are shown in Figure 6. Our results indicate that the convergence performance of AdaLomo is unaffected by the use or absence of gradient normalization. We attribute this to the grouped update normalization feature within AdaLomo. Avoiding the use of gradient normalization can eliminate the need for two backward passes, thus preventing computational redundancy during training.

## B  INSTRUCTION TUNING

### B.1  HYPER-PARAMETERS

Hyper-parameters used by different optimization methods and models for instruction-tuning are shown in Table 3.

### B.2  TEMPLATES

Templates used for instruction-tuning on Alpaca-GPT4 are shown in Table 4.

## C  FURTHER PRE-TRAINING

### C.1  HYPER-PARAMETERS

Hyper-parameters used for further pre-trianing are shown in Table 5.

Table 3: Hyper-parameters for instruction-tuning.

| | LLaMA-7B | | | | LLaMA-13B | | | |
| | LoRA | AdamW | LOMO | AdaLomo | LoRA | AdamW | LOMO | AdaLomo |
|---|---|---|---|---|---|---|---|---|
| Learning Rate | 3E-04 | 2E-05 | 1E-02 | 5E-04 | 3E-04 | 2E-05 | 1E-02 | 5E-04 |
| Batch Size | 128 | | | | | | | |
| Ecochs | 3 | | | | | | | |
| Warmup Steps | 0.03 * Total Steps | | | | | | | |
| | LLaMA-30B | | | | LLaMA-65B | | | |
| | LoRA | AdamW | LOMO | AdaLomo | LoRA | AdamW | LOMO | AdaLomo |
| Learning Rate | 3E-04 | 2E-05 | 1E-02 | 5E-04 | 3E-04 | 1E-05 | 1E-02 | 5E-04 |
| Batch Size | 128 | | | | | | | |
| Ecochs | 3 | | | | | | | |
| Warmup Steps | 0.03 * Total Steps | | | | | | | |

Table 4: Templates used for instruction-tuning.

**Template for entries with input**

Below is an instruction that describes a task, paired with an input that provides further context. Write a response that appropriately completes the request.

### Instruction:
{instruction}

### Input:
{input}

### Response:{response}

**Template for entries without input**

Below is an instruction that describes a task. Write a response that appropriately completes the request.

### Instruction:
{instruction}

### Response:{response}

## C.2 DATA SAMPLES

**Chinese**  A Chinese data sample is shown in Figure 7.

**Python code**  A Python code data sample is is shown in Figure 8.

## D MEMORY AND THROUGHPUT PROFILE

The hyper-parameters used to profile memory and throughput and the detailed results are shown in Table 6. The experiments are conducted on A800 with NVLink.

《孕产保健实用宝典》是2010年9月1日浙江科学技术出版社出版的图书，作者是杨惠民。
本书主要对胎教的含义和胎教的具体方法进行了详细的介绍。
##内容简介
如果说孩子是爱情结晶的话，那么面对父母的首要问题，就是如何提高和培养孩子的智能和如何使孩子健康的成长，顺利快乐的度过每一天。
在这个纷繁复杂的社会，人人都想让自己的孩子与众不同，虽然这一切主要受社会环境的影响，但其更大是受胎教的影响？
##目录
第一章
如何选择最佳怀孕时机
最佳的生育年龄
妊娠年龄过小有何缺点
妊娠年龄过大有何缺点
最佳身体条件
最佳的营养和受孕环境
第二章 ……

Figure 7: A Chinese data sample for further pre-training.

```python
# -*- coding: utf-8 -*-
# Generated by Django 1.11.1 on 2017-05-25 23:26
from __future__ import unicode_literals

from django.db import migrations, models

class Migration(migrations.Migration):

    dependencies = [
        ('podcast', '0006_podcast_language'),
    ]

    operations = [
        migrations.RemoveField(
            model_name='episode',
            name='file_ogg',
        ),
        migrations.AddField(
            model_name='episode',
            name='duration',
            field=models.DurationField(default='10'),
            preserve_default=False,
        ),
    ]
```

Figure 8: A Python code data sample for further pre-training.

Table 5: Hyper-parameters used for further pre-training.

| Method | AdamW | AdaLomo |
|---|---|---|
| Sequence Length | 2048 | |
| Learning Rate | 1E-05 | 3E-01 |
| Batch Size | 128 | |
| Warmup Steps | 0.03 * Total Steps | |

Table 6: Hyper-parameters and detailed results in memory and throughput profile.

| Model | Optimizer | GPUs | Micro Batch Size | Memory (GB) | Throughput (TGS) |
|---|---|---|---|---|---|
| LLaMA-7B | AdamW | 4 | 8 | 169.4 | 3169.4 |
| | Adafactor | | | 144.3 | 3169.5 |
| | LoRA | | | 70.6 | 3344.6 |
| | LOMO | | | 59.6 | 3228.2 |
| | AdaLomo | | | 59.6 | 2997.4 |
| LLaMA-30B | AdamW | 16 | 4 | 786.2 | 728.6 |
| | Adafactor | | | 665.0 | 726.5 |
| | LoRA | | | 303.7 | 811.6 |
| | LOMO | | | 264.3 | 669.1 |
| | AdaLomo | | | 272.8 | 589.0 |
| LLaMA-13B | AdamW | 8 | 4 | 320.7 | 1679.6 |
| | Adafactor | | | 272.3 | 1683.4 |
| | LoRA | | | 110.0 | 1829.8 |
| | LOMO | | | 94.4 | 1659.9 |
| | AdaLomo | | | 95.8 | 1456.3 |
| LLaMA-65B | AdamW | 32 | 2 | 1532.6 | 349.1 |
| | Adafactor | | | 1289.4 | 341.1 |
| | LoRA | | | 510.5 | 405.7 |
| | LOMO | | | 473.8 | 303.3 |
| | AdaLomo | | | 507.7 | 238.1 |

