# OpenReview forum: "AdaLomo: Low-memory Optimization with Adaptive Learning Rate"
_ICLR.cc/2024/Conference — ICLR 2024 Conference Withdrawn Submission_

### Official Review · Reviewer_VMbV · 2023-10-23

**Soundness:** 1 poor
**Presentation:** 2 fair
**Contribution:** 1 poor
**Rating:** 3
**Confidence:** 4

**Summary:**

This manuscript aims to provide a low-memory optimizer for LLMs.

**Strengths:**

* This manuscript studies a crucial problem in the field.
* This manuscript considers fine-tuning LLaMA with various model sizes

**Weaknesses:**

* The novelty of the AdaLomo is limited. It is very close to the design of Adafactor, while no performance comparison can be found in Table 2, Figure 2, and Figure 3.
* The convergence of AdaLomo has no theoretical guarantee.
* Some statements are weak, as no corresponding supports can be found. For example, in the sentence ``Through our ablation study on Adam, we found that its second-order moment estimation has a significantly greater impact on its convergence than the first-order moment estimation'', the ablation study refers to which table/figure?
* Some design choices have no justification. For example, why it is a good idea to consider $u_{t, i} / \max(1, RMS(u_{t, i})) \times \max(\epsilon, RMS(\theta_{t-1, i}))$ (though it is borrowed from AdaFactor)? Compared to other gradient clipping ideas, and gradient normalization techniques, why current design choice is a better idea? Some theoretical justifications or empirical evidence should be provided.
* LoRA is a parameter-efficient fine-tuning method, while AdamW and AdaLoMO are optimizers. It looks unfair to directly compare LoRA with AdamW and AdaLoMO as they are orthogonal.
* Some other baseline optimizers need to be considered. e.g., Lion [1] and Adan [2]; or even some other memory-efficient techniques should be evaluated with any other adaptive optimizers.

## Reference
[1] Symbolic Discovery of Optimization Algorithms

[2] Adan: Adaptive Nesterov Momentum Algorithm for Faster Optimizing Deep Models

**Questions:**

* What is the exact definition of ``SGD with variance''?
* What is the choice of optimizer used for LoRA?

---

### Official Review · Reviewer_t9K6 · 2023-10-27

**Soundness:** 2 fair
**Presentation:** 2 fair
**Contribution:** 2 fair
**Rating:** 3
**Confidence:** 3

**Summary:**

This paper presents the low-memory optimizer with adaptive learning rate (AdaLomo) for training large language models. The optimizer combines LOMO with the first/second-order moment estimations used in adaptive methods such as Adam. Computing the second-order moment estimation uses Non-negative matrix factorization. This paper shows numerically that AdaLomo is comparable with AdamW and reduces memory requirements.

**Strengths:**

The strength of this paper is to show numerically that the proposed method, AdaLomo, is comparable with AdamW and reduces memory requirements.

**Weaknesses:**

This paper numerically studies the performance of AdaLomo that can be obtained by combining the existing methods such as LOMO and Adam. It seems that AdaLomo has the best of both LOMO and Adam. This paper lacks theoretical explanations why AdaLomo performs better than the existing optimizers, such as LoRA, AdamW, and LOMO.

**Questions:**

In general, I am not very familiar with this topic. Although this paper studies mainly practical numerical comparisons, I think that it needs theoretical explanations and evidences to emphasize the usefulness of AdaLomo. In particular,

- There have been convergence analyses of Adam. Is it guaranteed theoretically that AdaLomo converges to a (local) minimizer and does not fall into the worst local minimizer (since Figure 1(b) shows that Adam decreases the value of the loss function more than SGD and SGD with momentum)?

Moreover, I have the following concern.
- The optimizers use that a batch size $b$ is 128. I do not know why the setting is appropriate. In intuition, the optimizers with a smaller batch size than 128 decrease the value of the loss function. Please compare the performance of using $b=128$ with the one of using smaller/larger batch sizes than $b = 128$. I would also like to check the performances of the optimizers using a cosine annealing step-size and a step-decay step-size. Could you compare the optimizers using constant step-sizes (Table 3) with the ones using cosine annealing step-sizes/step-decay step-sizes?

Typos and minor comments:
- Page 1, Line -9: identifie $\to$ identify
- Page 4, Line -14: What is "Adafactor"?
- Algorithm 1, Step 9: $s_{t,i}$ is not defined?
- Page 5, footnote 1: $\sum_{i=1}^{i=n} \to \sum_{i=1}^n$

---

### Official Review · Reviewer_6pvp · 2023-11-01

**Soundness:** 1 poor
**Presentation:** 1 poor
**Contribution:** 1 poor
**Rating:** 1
**Confidence:** 5

**Summary:**

This paper proposes the AdaLOMO optimizer for training large language models. Specifically, the work 1) takes Adam and removes the second moment buffer, 2) factors the second momentum buffer of each parameter group into a Rank 1 matrix (outer-product of two vectors) and re-scales the second moment update to have a similar magnitude to the inverse RMS of the previous iterations parameters in the case where the updates grow very large. Numerical experiments are provided with LLaMA instruction tuning, and further LLaMA pretraining.

**Strengths:**

### Significance
* This paper studies an important problem, namely how to improve the memory efficiency of optimizing large language models.

**Weaknesses:**

### Originality
* The proposed method is EXACTLY Algorithm 5 in Shazeer et al., 2018. The only difference is that the gradient of a layer is deallocated once it is used to compute the gradient of the next layer so as to save memory.
  * Adafactor does already propose using Adam with a factored second moment. Similarly, they remove the first moment buffer. The RMS normalization is also taken directly from Adafactor, where it is motivated as a strategy for preventing very large updates when using slow decay of the second order buffer in the absence of a first order buffer.

### Quality
Numerous mathematical issues in terms of clarity + a few minor mathematical errors which are probably typos.
* RMS on page 5 is incorrect, terms inside summation should be squared. However, $u$ is a matrix, so unclear what this computation involves.
* In equation 9 it should be $v_{t,i} = r_{t,i} c_{t, i}$
* In equation 10, unclear what $g_{t,i} / v_{t,i}$ involves, since $g_{t,i}$ and $v_{t,i}$ are matrices of size $m \times n$
* Used $\theta_{t,i}$ to refer to a “parameter” but this is described as matrix of size $m \times n$, should it is not just a single parameter.

**Questions:**

What are the novel contributions of AdaLOMO compared to Adafactor?

---

### Official Review · Reviewer_vpaW · 2023-11-02

**Soundness:** 3 good
**Presentation:** 3 good
**Contribution:** 2 fair
**Rating:** 6
**Confidence:** 3

**Summary:**

This paper proposes AdaLomo, a low-memory optimization method for large language models that provides an adaptive learning rate for each parameter while maintaining memory efficiency. The key ideas are using non-negative matrix factorization to estimate the second-order moment for the adaptive learning rate, and employing grouped update normalization to stabilize training. Experiments on instruction tuning and further pretraining of LLaMA models show AdaLomo achieves comparable results to AdamW while significantly reducing memory requirements.

**Strengths:**

* Tackles the important challenge of reducing memory footprint for large LM training. The motivation is well articulated.
* Empirically analyzes differences between SGD, Adam and LOMO highlighting the role of second-order moments. Provides insight.
* AdaLomo integrates sensible ideas - NMF for second-order moment estimation and grouped update normalization - to offer adaptive learning rates with low memory.
* Experiments cover instruction tuning and further pretraining with solid results on par with AdamW and LoRA. Reduced memory requirements demonstrated.

**Weaknesses:**

* The main ideas borrowed from prior work like Adafactor and grouped normalization limit novelty. Contributions appear incremental.
* The two core components of AdaLomo are the use of non-negative matrix factorization (NMF) for estimating the second-order moment and the grouped update normalization. However, the paper does not contain ablation studies to directly demonstrate the benefits of each component.
* Convergence plots during pretraining could be insightful to compare optimization behavior.
* Lack of comparisons to related memory efficient methods like SM3, ZeRO, and 8-bit optimizer.

**Questions:**

* Have you experimented with other ways to estimate second-order moments besides NMF? How did they compare in terms of memory and performance?
* How sensitive is AdaLomo to the hyperparameters? Any guidelines for settings based on model architecture or task?